# Assessment of Water Quality and Thermal Stress for an Artificial Fish Shelter in an Urban Small Pond during Early Summer

**Chang Hyuk Ahn [1], Saeromi Lee [1], Ho Myeon Song [1], Jae Roh Park [1,*] and Jin Chul Joo [2,*]**

[1] Department of Land, Water and Environment Research, Korea Institute of Civil Engineering and Building Technology, Goyang-Si 10223, Korea; chahn@kict.re.kr (C.H.A.); saeromi@kict.re.kr (S.L.); hmsong@kict.re.kr (H.M.S.)

[2] Department of Civil & Environmental Engineering, Hanbat National University, Daejeon City 34158, Korea

* Correspondence: jrpark@kict.re.kr (J.R.P.); jincjoo@hanbat.ac.kr (J.C.J.); Tel.: +82-31-910-0302 (J.R.P.); +82-42-821-1264 (J.C.J.); Fax: +82-31-910-0291 (J.R.P.); +82-42-821-1476 (J.C.J.)

**Abstract:** This study evaluated water quality variations in an artificial deep pool (ADP), which is an underground artificial structure built in a shallow pond as a fish shelter. The water temperature, pH, dissolved oxygen (DO), and electrical conductivity (EC) were measured on an hourly basis in the open space and inside the ADP, and a phenomenological study was performed, dividing seasons into normal and rainy seasons and environments into stagnant and circulating conditions. The results showed that the water quality parameters inside the ADP exhibit lower fluctuations and diurnal variations compared with the open space. On average, the water temperature inside the ADP is lower than outside it by 1.7–3.7 °C in stagnant conditions, and by 0.6–0.7 °C in circulating conditions during early summer. Thermal stratification occurs inside the ADP but is temporarily disturbed due to the mixing from the forced circulation and the rainwater input through rainfall events. The ADP provided a constant and optimal water temperature for living and spawning for bitterling (i.e., 15.0–21.0 °C), which dominated in experimental pond during spring to summer. Most importantly, the ADP was able to significantly reduce the thermal stress of the fish in the study site, and as a result, the bitterling, a cool water fish species, could successfully become dominant. Finally, the deployment of the ADP appears to provide a practical alternative for effective fishery resources management to improve species diversity and fish communities in an artificial freshwater ecosystem (garden pond, park pond, other artificial wetlands, etc.).

**Keywords:** artificial deep pool (ADP); water temperature; thermal stress; diurnal variation; fish shelter

## 1. Introduction

Fish are exposed to many stress factors throughout their lifetime [1]. As water is the sole living space for fish, the aquatic environment is a directly limiting factor that influences the survival of fish [2,3]. In particular, water quality parameters such as water temperature, pH, and DO, whose optimal ranges differ from one fish species to another, determine the changes in the physiology of fish [4–7].

In particular, fish are very sensitive to water temperature [6,8]. Excessively high water temperatures or high diurnal variations cause thermal stress in fish, and this can be aggravated through accumulation [9]. As such, fish are exposed to continuous cycles of stress and de-stress depending on the rise and fall of the water temperature [10]. Cheung et al. also reported that approximately 24% of fish species worldwide face the threat of extinction as a result of increases in

water temperature related to global warming [11]. As such, various survival strategies are required to mitigate the impact of accumulated thermal stress on fish.

In this respect, summer, the season of rising temperatures, can be a harsh period for fish. For example, an excessive increase in temperature can induce numerous physiological changes in the body of fish. Previous studies have shown that fish exposed to high temperatures have protein damage, hormonal changes, and high mortality due to thermal stress [12,13]. Therefore, adequate water management focused on water temperature is closely related to the health of the fish, so appropriate preparation is required, especially in the season when water temperature is rising.

As mentioned, since excessive rises in water temperature in aquatic environments can be detrimental to the reproduction and growth of fish, there has been ongoing research into the functions of fish shelters [14–22]. Previous studies have found that fish shelters can be used to prevent untimely predator encounters [14,15], to enable the survival of physical disturbance events such as floods or droughts, and to maintain or increase fish populations through enhancing survival rates [19]. However, most studies have emphasized the complexities of fish shelter structures, and have focused on the vulnerability of prey fish. For this reason, a phenomenological approach is needed to evaluate the function of a fish shelter in mitigating the impact of accumulated thermal stress.

Recently, a large number of water spaces have been introduced into urban areas and gardens to improve ecological functions [23,24]. However, fish are easily exposed to thermal stress due to insufficient water volume and shallow water depth. In these circumstances, an additional method of securing depth, such as a deep pool, can be used as an alternative. Deep pools in their natural state are generally known to mitigate elevated water temperatures due to their geometric structure and depth, which can contribute to increased survival rates and healthy habitats for fish [20,22].

The ADP is developed with the aim of providing shelter to improve fish survival and to mitigate thermal stress in shallow ponds reflecting the above considerations. The ADP is derived from traditional pools that were used in the paddy fields in the Republic of Korea, and has many advantages in terms of an ecological habitat for freshwater organisms. The basic structure of the ADP is a cuboid consisting of a cover and a main body; the main body is the space of the fish shelter, and the cover consists of holes (diameter 0.2 m) and lateral entrances (height 0.2 m) allowing fish to move. The cover is a pathway for fish to evacuate into the ADP, and prevents people from falling in while managing the ponds. In the previous study, the ADP is an underground structure that secures a shelter space for fish to escape from adverse conditions during the dry season [19]. Also, ADP was verified to be both an effective fish shelter and a natural habitat area for endangered fish species [19]. Therefore, in this study, we tried to investigate water quality characteristics and to verify the ability to mitigate thermal stress of fish when ADP was applied to a shallow pond.

To evaluate the function of a fish shelter in mitigating the impact of accumulated thermal stress, the water quality inside the ADP was investigated for two years during the period from early summer when thermal stress tends to rise in fish. The specific objectives of this study were (1) to confirm the characteristics of the water quality inside the ADP through a phenomenological approach, (2) to evaluate utilization possibilities for the ADP buried at the bottom of a shallow pond, and (3) to perform a quantitative evaluation of the thermal stress that can affect the ecological health for fish.

## 2. Materials and Methods

### 2.1. Study Site

The mesocosm experiments were conducted in a pond at the Korea Institute of Civil Engineering and Building Technology (KICT) in Gyeonggi-do, Republic of Korea. As shown in Figure 1, the specifications of the pond (the study site) are as follows: surface area (110 $m^2$), average water depth (about 0.5 m), and maximum water depth (about 0.7 m). Both gravel (diameter ≤ 60 mm) and sand (diameter ≤ 2 mm) were used as bed materials, and a bentonite liner (5 cm) was used as an impermeable layer. The water level was kept constant throughout the year using both harvested rainwater and tap water, with a water level sensor and underwater pump installed at the water inflow point. This pond is basically a lentic system, but when the water level drops, the underwater

pump supplies the necessary water flow. In addition, when a large amount of water flow is supplied by rainfall, the excess is discharged to the outlet. Therefore, even though the pond is small, it can always have a constant water volume and depth.

The ADP with dimensions of 1.5 m (L) × 1.5 m (W) × 1.5 m (H) was constructed with cement-zero concrete. Holes were perforated on the cover (diameter 0.2 m), and four sides (height 0.2 m) were open to allow fish to enter the ADP. Basalt (diameter ≤ 250 mm) was used to make the entrances appear natural to fish (see Figure 1). An underwater pump (IP-217, Hanil Industrial Co. Ltd., Seoul, Republic of Korea) was installed inside the ADP at a water depth of 2.0 m, and was operated at a rate of 50 L min$^{-1}$ with a regular on–off interval of 30 min. To evaluate the effects of both stagnation and circulation of pond water on the water quality inside and outside the ADP, the underwater pump was not operated in 2012, but was operated in 2013. These attempts help us to compare the thermal stress of fish according to the application type of the ADP.

The significant biota of the pond includes fringed waterlily (*Nymphoides peltata*) and bitterling (*Rhodeus uyekii*). While there were no noticeable changes in the pond specifications and biota, the covered water surface of fringed waterlily increased from approximately 15% in May 2012 to 80% in 2013.

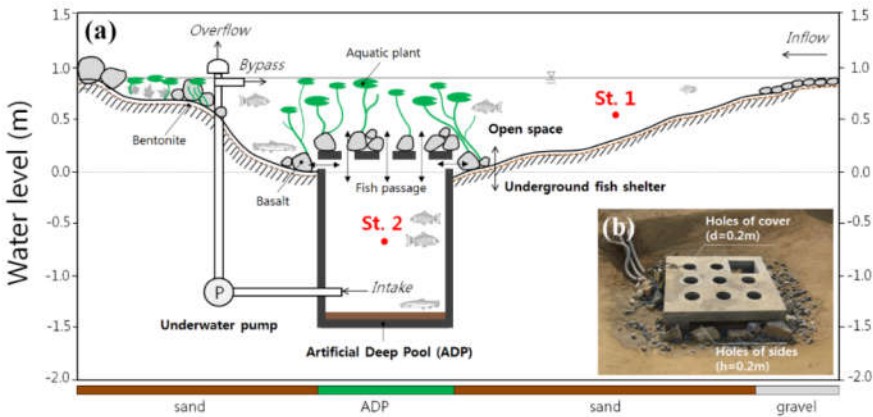

**Figure 1.** (**a**) Conceptual view of the study site. The y-axis is the water level and the x-axis is the bottom materials (not to scale). The intake point of the underwater pump is at a water depth of 1.0 m inside the ADP. The water quality monitoring sensors are at a water depth of 0.4 m in open space (St. 1) and 1.5 m inside the ADP (St. 2). (**b**) Pictorial view of constructed ADP in the bottom of the pond. The main body (fish shelter) is buried underground and fish can vertically or horizontally move through the holes (fish passages) in the cover and sides.

*2.2. Water Quality and Fish Monitoring Analysis*

Two water quality measurement points were designated at St. 1 for the open space and at St. 2 inside the ADP (see Figures 1 and 2). The water quality monitoring sensors (XLM6000, YSI, Yellow Springs, OH, USA) were installed at depths of 0.4 m (representing the open space) and 1.0 m (representing the inside the ADP) for St. 1 and St. 2, respectively, to measure the water temperature, pH, DO, and EC. Measurements were conducted remotely every hour for 132 days from May 21 to July 25 in 2012 and 2013. Since the monsoon season in the Republic of Korea generally begins in June and lasts for approximately one month [25], the measured data were classified into normal and rainy season. The depth-dependent data inside the ADP were generated by portable devices (550A, 63; YSI) during a non-pump action period or with the underwater pump off for a while.

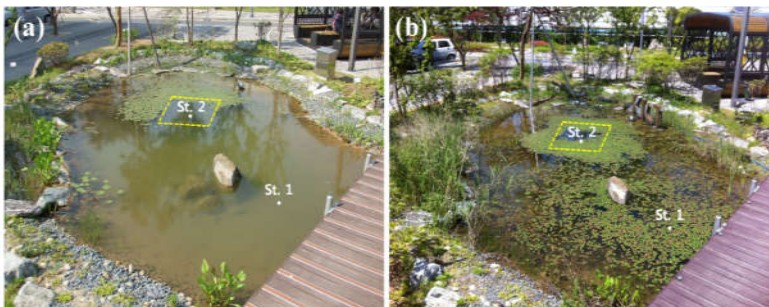

**Figure 2.** Installation of the ADP at the study site. St. 1 and St. 2 are placed in the areas of open space and ADP, respectively. Photo (**a**) is a view of the pond in May 2012 and photo (**b**) is a view of the pond in May 2013.

The DO saturation was calculated using Equation (1) proposed by Colt [26]. The barometric pressure (BP) was assumed to be 760 mmHg (i.e., 1 atm), and the Bunsen coefficient of oxygen ($B_{O_2}$) represents the temperature-dependent level of dissolved oxygen. The vapor pressure ($P_{H_2O}$) was obtained through substituting the temperature-dependent index,

$$\text{DO}_{saturation}\ (\%) = \left[\frac{0.5318\left(\frac{\text{DO}}{B_{O_2}}\right)}{0.20946\left(\text{BP} - P_{H_2O}\right)}\right] \times 100 \tag{1}$$

Here, BP denotes the local barometric pressure (mmHg), DO denotes the dissolved oxygen concentration (mg L$^{-1}$), $B_{O_2}$ denotes the Bunsen coefficient of oxygen as a function of temperature, and $P_{H_2O}$ denotes the vapor pressure of water as a function of the temperature (mmHg).

Samples for physicochemical water quality were collected using a Van Dorn sampler [27]. A polyethylene bottle with 2 L of the water sample was transported to the laboratory, and the physicochemical parameters of the samples were analyzed for turbidity, SS, TOC, DOC, BOD, COD$_{Mn}$, TN, NH$_3$, NO$_3{}^-$, TP, PO$_4{}^{3-}$, and Chl-*a* using standard methods [28]. Rainfall data were obtained from the database from the Gimpo Airport of the Korea Aviation Meteorological Agency (KAMA) about 12 km from the study site.

The fish were monitored four times over the study period using capture per unit effort (CPUE) method by kick net (mesh 3 mm × 3 mm) in the study site. The investigation site was the entire pond and the blocking net (mesh 3 mm × 3 mm) was placed on the cover of the ADP prior to fish monitoring to minimize the interference of the fish shelter. Relative abundance result was calculated on average based on the captured fish population.

*2.3. Statistical Analysis*

Based on the measurement data, a statistical analysis of the water quality parameters observed at between St. 1 and St. 2 was performed, and their interactions using a two-way ANOVA was evaluated. A two-way ANOVA test can determine the significant difference between the mean of the dependent variable for multiple groups, and identify significant interactions among the variations [3,29].

*2.4. Thermal Stress Analysis*

The thermal stress model described by Bevelhimer and Bennett consists of factors with accumulation and recovery according to water temperature [10]. In this model, the magnitude and duration of high water temperature exposure, the stress recovery during periods of reduced water temperature, and fish characteristic of threshold (or final preference temperature) are included. Therefore, the water temperature data as a main factor of this study is expressed as thermal stress based on the following equations.

$$\text{Thermal stress (t)} = \text{Thermal stress (t} - dt) + (\text{Accumulation} - \text{recovery})dt \tag{2}$$

$$\text{Accumulation} = \text{Ambient water temperature} - \text{Threshold} \qquad (3)$$

$$\text{Recovery} = (\text{Threshold} - \text{Ambient water temperature})Z \qquad (4)$$

Here, t denotes current time, *dt* denotes size of the time step, accumulation denotes index of the thermal stress accumulation, recovery denotes index of the thermal stress recovery, and *Z* is the factor delaying the rate of recovery. In this study, we assume recovery occurs at a rate of 25% that at which it accumulates [10].

Bevelhimer and Bennett explained that the model assumes that thermal stress accumulation occurs above a threshold water temperature at a rate dependent on the degree to which the threshold is exceeded [10]. The model also includes thermal stress recovery when temperatures drop below the threshold temperature, as in systems with large daily variation for fish species. Based on this point, we used Equations (2) to (4) to evaluate ADP's thermal stress reduction performance.

## 3. Results and Discussions

### 3.1. Water Temperature, pH, DO, and EC

The water temperature, pH, DO and EC measured by monitoring sensors in 2012 (i.e., under stagnant conditions) and 2013 (i.e., under circulation conditions) are summarized in Figure 3. For the period of stagnant conditions in 2012, as displayed in Figure 3a–d, a comparison of water quality analyses between St. 1 and St. 2 through two-way ANOVA revealed considerable season-dependent differences in the water temperature, pH, DO, and EC ($p < 0.05$). In this period, the values for the water temperature, pH, and DO at St. 2 were generally lower than those at St. 1, presumably due to the shading effect of the ADP.

On the other hand, the differences in the ambient environment between St. 1 and St. 2 during the rainy season were as distinct as those during the normal season. Considering that total precipitation levels during the normal and rainy seasons were 1 mm and 551 mm, respectively, the negligible differences between St. 1 and St. 2 during the rainy season can be attributed to the rainfall-induced mixing effect.

Different trends in water quality parameters within the ADP were observed under the stagnant (see Figure 3a–d) and the circulation conditions (see Figure 3e–h). In other words, the stagnant and circulation conditions in this study are based on the ADP flow conditions by applying the underwater pump. While the water temperature, pH, and DO exhibited markedly higher variations at St. 1 under the stagnant conditions, those at St. 2 exhibited relatively stable conditions, temporarily exhibiting the same levels as those outside the ADP during heavy rainfall events. However, similar values were measured at St. 1 and St. 2 under the circulation conditions, irrespective of the effect of heavy rainfalls (see Figure 3e–h). This implies that the mixing effect was sufficiently robust to homogenize the water quality in both ADP and open space. The EC was found to clearly decrease with increases in the dilution of the water body induced by rainwater (see Figure 3d,h).

Through the analyses, it was found that the differences in water quality parameters were greater under the stagnant condition compared with under the circulation condition. Under the stagnant condition, the difference [i.e., (St. 1 − St. 2)*dt*] in the water temperature was considerably more distinct, with the maximum 3.9–6.0 °C and the minimum 0.1–1.4 °C, while the average water temperature at St. 2 was lower than St. 1 by 1.7–3.7 °C (see Figure 3a and Table 1). This indicates that an ADP can function as a space for fish to recover from thermal stress.

The water temperature directly influences the physiological features (i.e., metabolic demands, digestion rates, and assimilation efficiencies) of fish [7,30]. Also, high water temperatures can cause thermal stress in fish [31], and thus fish require higher metabolic rates, which make fish vulnerable to disease and suppresses their growth [32]. Since the ADP was proven to provide constant and optimal water temperature for living and spawning (i.e., 15.0–21.0 °C) for the bitterling that dominated the experimental pond [33–35], it can serve as a fish shelter for fish vulnerable to heat during periods of high water temperatures.

As with water temperature, the other basic differences [i.e., (St. 1 − St. 2)*dt*] in water quality were more distinct during the stagnant condition, where the average pH ranged from 0.1–0.2, and average DO from 2.7–3.6 mg L$^{-1}$. The positive values of the pH and DO indirectly imply higher photosynthesis efficiency in the open space compared to inside the ADP. But EC did not show a significant spatial difference in St. 1 and St. 2 (see Figure 3b,c and Table 1).

Under the circulation condition, the difference [i.e., (St. 1 − St. 2)*dt*] in the water quality parameters was less prominent, with the maximum 0.9–1.0 °C, minimum 0.3–0.5 °C and the average water temperature at St. 2 was lower than St. 1 by 0.6–0.7 °C (see Figure 3e and Table 2). Also, neither pH nor DO were significantly different inside the ADP compared to in the open space (see Figure 3f,g). These results can be attributed to the circulation and pumping-induced photosynthesis of the aquatic plants and attached algae densely distributed on the outside of the ADP cover.

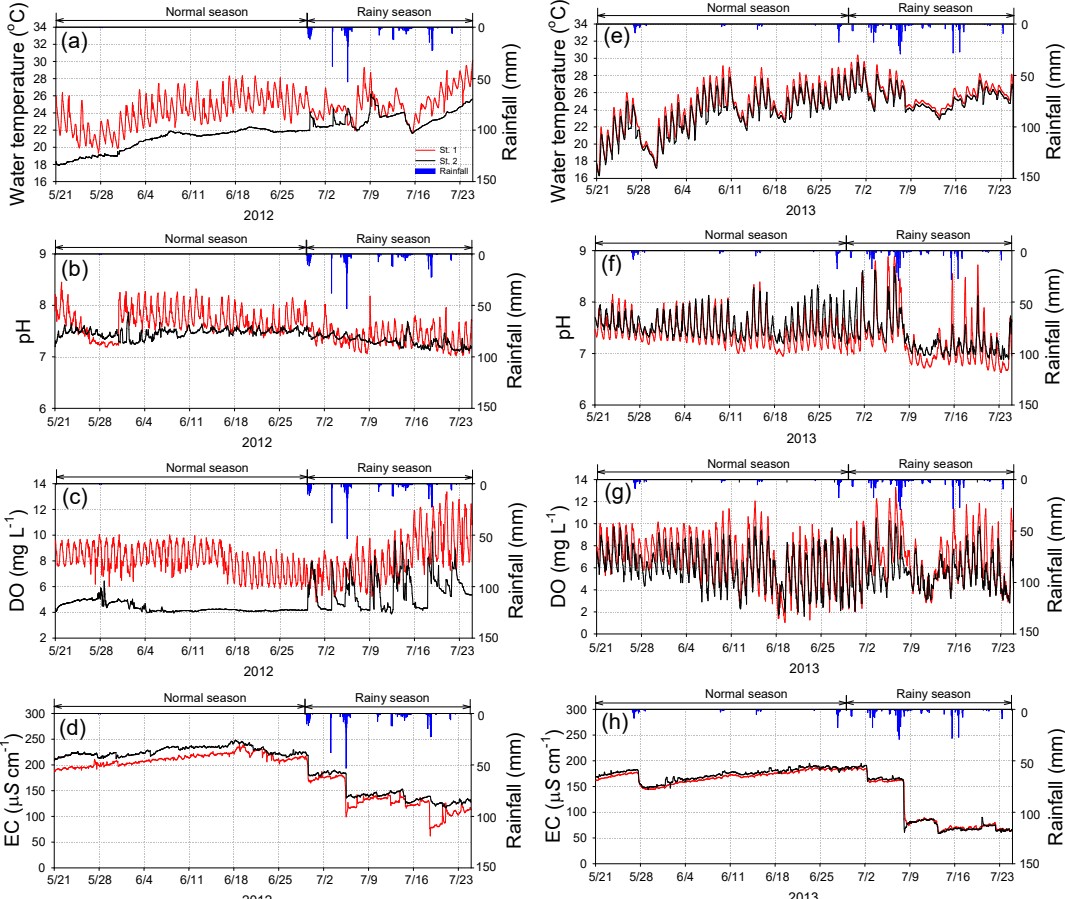

**Figure 3.** Variations in the water temperature, pH, DO, and EC values of St. 1 and St. 2 during the normal and rainy seasons under stagnant (**a**–**d**; 2012) and circulation (**e**–**h**; 2013) conditions. The water depth kept constant, regardless of time order, and the maximum water depth was 0.69 ± 0.01 m in 2012 and 0.69 ± 0.03 m in 2013 (*n* = 7).

To verify the occurrence of oxic/anoxic layers and thermal stratification inside the ADP, the water temperature, pH, DO, and EC were observed at different depths. As depicted in Figure 4, under stagnant conditions, no significant depth-dependent differences in pH and EC were observed (see Figure 4b,d), whereas both water temperature and DO rapidly decreased at the depth intervals between 0.8 and 1.5 m, indicating that an anoxic layer existed in these water bottoms. In particular, a considerable thermal stratification in average water temperature and DO was observed between depths of 0.8 m and 1.5 m (see Figure 4a,c).

**Table 1.** Measurement data of the water quality parameters during normal and rainy seasons under stagnant conditions (2012).

| Description | Normal Season | | | | | | Rainy Season | | | | | |
|---|---|---|---|---|---|---|---|---|---|---|---|---|
| | St. 1 | | | St. 2 | | | St. 1 | | | St. 2 | | |
| | Mean ± S.D. | Max. | Min. | Mean ± S.D. | Max. | Min. | Mean ± S.D. | Max. | Min. | Mean ± S.D. | Max. | Min. |
| Temperature (°C) [a] | 24.1 ± 2.0 | 28.4 | 19.3 | 20.4 ± 1.5 | 22.4 | 17.9 | 25.1 ± 1.7 | 30.1 | 21.7 | 23.4 ± 1.0 | 26.2 | 21.6 |
| pH [a] | 7.7 ± 0.3 | 8.5 | 7.2 | 7.5 ± 0.1 | 7.9 | 7.2 | 7.4 ± 0.2 | 8.2 | 7.0 | 7.3 ± 0.1 | 7.7 | 7.1 |
| DO (mg L$^{-1}$) [a] | 8.1 ± 1.0 | 10.1 | 5.8 | 4.5 ± 0.4 | 6.4 | 4.0 | 8.3 ± 1.8 | 13.4 | 4.8 | 5.6 ± 1.3 | 10.3 | 3.9 |
| DO saturation (%) [c] | 96.2 ± 12.1 | 123.6 | 68.7 | 49.2 ± 3.3 | 69.8 | 44.5 | 101.1 ± 23.7 | 169.7 | 56.4 | 66.1 ± 15.5 | 120.1 | 46.0 |
| EC (µS cm$^{-1}$) [a] | 211 ± 11 | 239 | 187 | 225 ± 8 | 248 | 208 | 133 ± 29 | 217 | 62 | 148 ± 24 | 223 | 117 |
| Turbidity (NTU) [b] | 8.8 ± 2.8 | 12.0 | 6.5 | 7.3 ± 1.2 | 8.3 | 6.0 | 6.0 ± 0.2 | 6.2 | 5.8 | 6.5 ± 0.6 | 7.2 | 6.1 |
| SS (mg L$^{-1}$) [b] | 9.4 ± 1.9 | 11.5 | 7.8 | 5.1 ± 0.7 | 5.8 | 4.5 | 8.1 ± 0.3 | 8.4 | 7.8 | 8.3 ± 0.4 | 8.7 | 7.9 |
| TOC (mg L$^{-1}$) [b] | 1.77 ± 0.14 | 1.79 | 1.73 | 1.75 ± 0.14 | 1.78 | 1.73 | 2.04 ± 0.22 | 2.42 | 1.75 | 1.99 ± 0.15 | 2.28 | 1.81 |
| DOC (mg L$^{-1}$) [b] | 1.69 ± 0.15 | 1.74 | 1.61 | 1.71 ± 0.15 | 1.75 | 1.67 | 1.85 ± 0.11 | 2.05 | 1.69 | 1.91 ± 0.18 | 2.25 | 1.71 |
| BOD (mg L$^{-1}$) [b] | 3.1 ± 0.5 | 3.5 | 2.6 | 3.1 ± 0.3 | 3.3 | 2.8 | 3.3 ± 0.2 | 3.5 | 3.2 | 3.1 ± 0.2 | 3.3 | 3.0 |
| COD$_{Mn}$ (mg L$^{-1}$) [b] | 2.8 ± 0.3 | 3.1 | 2.4 | 2.8 ± 0.3 | 3.1 | 2.3 | 2.6 ± 0.2 | 2.9 | 2.4 | 2.7 ± 0.1 | 2.9 | 2.5 |
| TN (mg L$^{-1}$) [b] | 1.2 ± 0.1 | 1.3 | 1.1 | 1.2 ± 0.1 | 1.3 | 1.1 | 1.3 ± 0.1 | 1.4 | 1.2 | 1.3 ± 0.1 | 1.4 | 1.2 |
| NH$_3$ (µg L$^{-1}$) [b] | 159 ± 7 | 166 | 152 | 167 ± 10 | 178 | 158 | 160 ± 10 | 170 | 150 | 170 ± 13 | 182 | 156 |
| NO$_3^-$ (µg L$^{-1}$) [b] | 1043 ± 25 | 1070 | 1020 | 1039 ± 22 | 1063 | 1021 | 1033 ± 21 | 1050 | 1010 | 1034 ± 14 | 1048 | 1.021 |
| TP (µg L$^{-1}$) [b] | 36 ± 3 | 40 | 34 | 40 ± 5 | 45 | 36 | 42 ± 4 | 47 | 39 | 43 ± 2 | 45 | 42 |
| PO$_4^{3-}$ (µg L$^{-1}$) [b] | 22 ± 3 | 25 | 19 | 24 ± 4 | 29 | 21 | 16 ± 2 | 18 | 14 | 18 ± 4 | 21 | 14 |
| Chl-*a* (µg L$^{-1}$) [b] | 8.5 ± 1.1 | 9.6 | 7.5 | 1.1 ± 0.4 | 1.5 | 0.8 | 13.5 ± 1.2 | 14.7 | 12.4 | 0.7 ± 0.6 | 1.2 | ND [d] |

[a] Water quality monitoring device; [b] Laboratory analysis; [c] DO saturation equation [26]; [d] ND: no data.

**Table 2.** Measurement data of the water quality parameters during normal and rainy seasons under the circulation conditions (2013).

| Description | Normal Season | | | | | | Rainy Season | | | | | |
|---|---|---|---|---|---|---|---|---|---|---|---|---|
| | St. 1 | | | St. 2 | | | St. 1 | | | St. 2 | | |
| | Mean ± S.D. | Max. | Min. | Mean ± S.D. | Max. | Min. | Mean ± S.D. | Max. | Min. | Mean ± S.D. | Max. | Min. |
| Temperature (°C) [a] | 24.0 ± 2.7 | 29.1 | 16.6 | 23.3 ± 2.5 | 28.1 | 16.3 | 26.0 ± 1.4 | 30.4 | 23.3 | 25.4 ± 1.3 | 29.5 | 22.8 |
| pH [a] | 7.4 ± 0.3 | 8.4 | 7.0 | 7.6 ± 0.2 | 8.3 | 7.2 | 7.2 ± 0.5 | 8.9 | 6.6 | 7.3 ± 0.4 | 8.7 | 6.9 |
| DO (mg L$^{-1}$) [a] | 6.9 ± 2.2 | 12.1 | 1.0 | 6.0 ± 1.7 | 9.7 | 1.6 | 6.7 ± 2.3 | 13.3 | 2.0 | 5.6 ± 1.6 | 10.5 | 2.1 |
| DO saturation (%) [c] | 81.7 ± 27.1 | 154.0 | 12.0 | 70.3 ± 20.8 | 122.7 | 19.3 | 85.1 ± 30.9 | 176.6 | 29.0 | 68.3 ± 20.8 | 133.3 | 16.6 |
| EC (μS cm$^{-1}$) [a] | 169 ± 11 | 187 | 145 | 174 ± 11 | 195 | 144 | 107 ± 47 | 189 | 60 | 107 ± 49 | 195 | 59 |
| Turbidity (NTU) [b] | 8.7 ± 1.7 | 12.0 | 6.5 | 8.0 ± 1.0 | 8.9 | 6.0 | 7.8 ± 1.8 | 10.1 | 5.8 | 8.8 ± 2.5 | 12.5 | 6.1 |
| SS (mg L$^{-1}$) [b] | 7.7 ± 2.1 | 11.5 | 5.1 | 5.6 ± 0.6 | 6.3 | 4.5 | 9.5 ± 1.7 | 12.1 | 7.8 | 10.0 ± 1.9 | 13.1 | 7.9 |
| TOC (mg L$^{-1}$) [b] | 2.20 ± 0.11 | 2.40 | 2.04 | 2.50 ± 0.18 | 2.76 | 2.22 | 2.25 ± 0.14 | 2.51 | 2.05 | 2.38 ± 0.14 | 2.54 | 2.10 |
| DOC (mg L$^{-1}$) [b] | 2.16 ± 0.11 | 2.35 | 1.97 | 2.45 ± 0.18 | 2.75 | 2.19 | 2.16 ± 0.12 | 2.35 | 1.97 | 2.29 ± 0.12 | 2.50 | 2.07 |
| BOD (mg L$^{-1}$) [b] | 3.2 ± 0.3 | 3.5 | 2.6 | 3.2 ± 0.2 | 3.3 | 2.8 | 3.2 ± 0.2 | 3.5 | 3.0 | 3.2 ± 0.1 | 3.3 | 3.0 |
| COD$_{Mn}$ (mg L$^{-1}$) [b] | 3.6 ± 0.1 | 3.8 | 3.4 | 3.5 ± 0.1 | 3.7 | 3.3 | 3.4 ± 0.2 | 3.8 | 3.2 | 3.4 ± 0.1 | 3.6 | 3.3 |
| TN (mg L$^{-1}$) [b] | 1.2 ± 0.1 | 1.3 | 1.1 | 1.2 ± 0.1 | 1.3 | 1.1 | 1.2 ± 0.1 | 1.4 | 1.0 | 1.2 ± 0.1 | 1.4 | 1.1 |
| NH$_3$ (μg L$^{-1}$) [b] | 146 ± 15 | 166 | 124 | 150 ± 19 | 178 | 127 | 138 ± 23 | 170 | 109 | 146 ± 26 | 182 | 118 |
| NO$_3^-$ (μg L$^{-1}$) [b] | 1048 ± 18 | 1070 | 1020 | 720 ± 437 | 1063 | 102 | 1.011 ± 55 | 1051 | 892 | 986 ± 69 | 1048 | 850 |
| TP (μg L$^{-1}$) [b] | 33 ± 4 | 40 | 27 | 35 ± 5 | 45 | 29 | 35 ± 8 | 47 | 26 | 37 ± 7 | 45 | 29 |
| PO$_4^{3-}$ (μg L$^{-1}$) [b] | 24 ± 3 | 28 | 19 | 25 ± 3 | 29 | 21 | 19 ± 3 | 22 | 14 | 21 ± 4 | 25 | 14 |
| Chl-*a* (μg L$^{-1}$) [b] | 10.4 ± 2.6 | 15.1 | 7.5 | 3.3 ± 2.2 | 6.1 | 0.8 | 14.5 ± 1.5 | 16.6 | 12.4 | 3.5 ± 2.9 | 7.1 | ND [d] |

[a] Water quality monitoring device; [b] Laboratory analysis; [c] DO saturation equation [26]; [d] ND: no data.

On the other hand, no significant depth-dependent differences were observed under the circulation conditions (see Figure 4e–h). The fact that there was no variation according to the water depth indicated that the circulation ensured relatively homogenous mixing of the water body inside the ADP due to the current generated by the underwater pump. Thus, thermal stratification occurred mainly in the stagnant condition, but not in the circulation condition.

The depth-dependent concentration range of the DO gradient was estimated to be 3.8–8.3 mg L$^{-1}$, and photosynthesis was limited to a depth of about 1.2 m (0.5 m below the ADP cover). In a previous study, it was found that the DO concentration required by fish is ≥5.0 mg L$^{-1}$ [36], but there are species that can adapt to lower concentrations (cool water fish ≥3.0 mg L$^{-1}$; warm water fish ≥2.5 mg L$^{-1}$) [37,38]. Thus, a higher distribution of DO-sensitive species and juveniles is expected beneath the cover, whereas a higher distribution of DO-non-sensitive species and adult fish is expected near the bottom.

Also, as the water depth increased, the decrease in the water temperature became more pronounced. Overall, the thermal stratification observed in the ADP suggested that different environmental conditions were found depending on the depth. Thus, in terms of water temperature and DO, the ADP can provide various aquaculture environments suitable for different fish species.

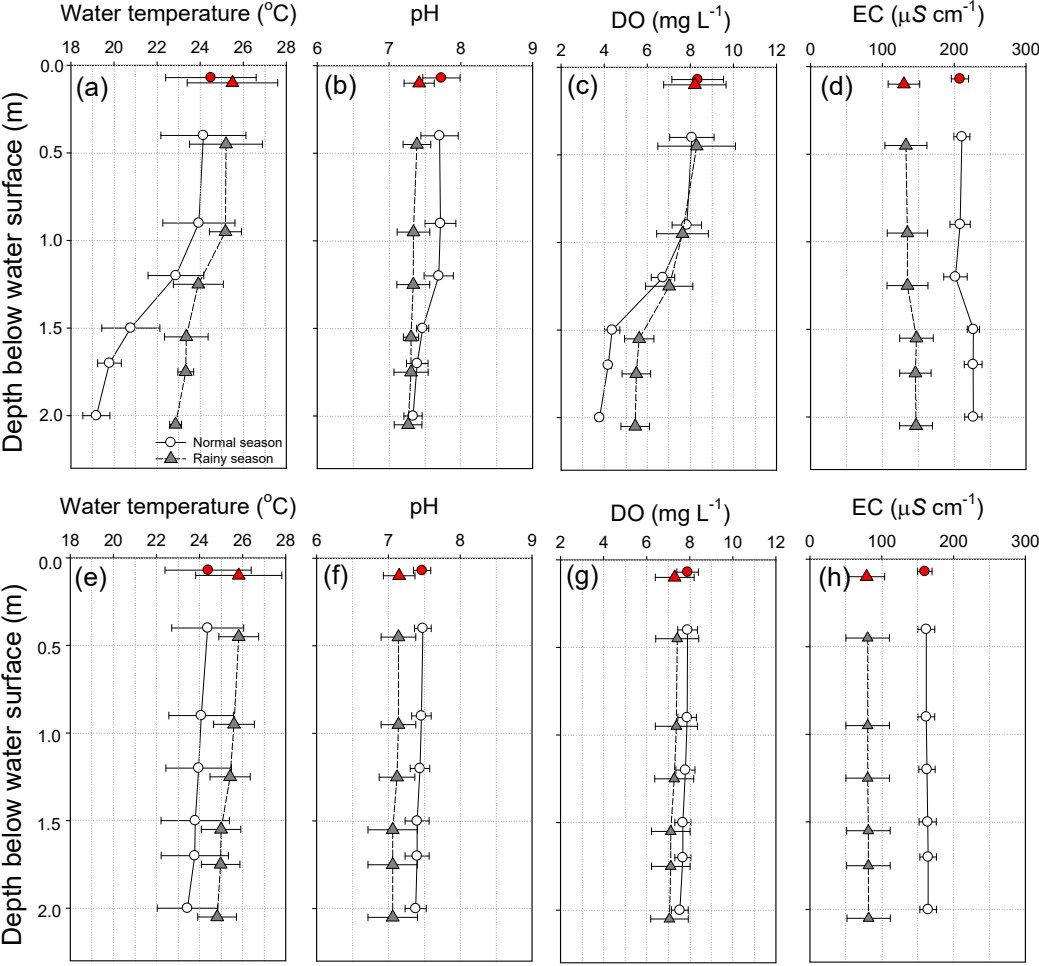

**Figure 4.** Vertical variations in the water temperature, pH, DO, and EC values inside the ADP under the stagnant (**a–d**; 2012) and the circulation (**e–h**; 2013) conditions by the underwater pump (*n* = 10). The red dots represent 0.1 m depth of water in the open space.

## 3.2. Physicochemical Water Quality

The results from the physicochemical water quality analyses performed in 2012 and 2013 were summarized in Tables 1 and 2, respectively. Overall, the physicochemical water quality in 2013 was slightly higher than in 2012 in terms of seasonality, but there was no significant difference between 2012 and 2013 in terms of the physicochemical water quality.

Nevertheless, there was a delicate difference in water quality between normal and rainy season in stagnant conditions. During the rainy season, greater values of both TOC and DOC, by 8.6% and 13.2%, respectively, were observed than in the normal season (see Table 1). This increase in TOC and DOC values during the rainy season is associated with the introduction of rainwater from the surface and suspended organic carbon from the sediments due to the external agitation of the pond. Unlike St. 1, a large part of St. 2 was shielded from sunlight, and both respiration and decay were expected to be dominant over photosynthesis. This phenomenon was reflected in the significant difference in the photosynthesis performance parameter (i.e., Chl-*a*) between St. 1 and St. 2 (8.5–13.5 and 0.7–1.1 µg L$^{-1}$, respectively).

The physicochemical water quality in the circulation condition was not significantly different from the stagnation condition. However, the concentrations of both organic and inorganic matters at St. 1 increased relative to those at St. 2. Compared with those values in 2012, TOC and DOC increased by 9.3–30.0% and 14.4–30.2%, respectively, presumably due to the increased volume of the fringed waterlily community, leading to increased detritus sources and attached algae. Chl-*a* was in the range of 3.3–16.6 µg L$^{-1}$, exhibiting a slight increase compared with levels in 2012 (see Table 2).

Based on these results, it was concluded that seasonal physicochemical water quality was not significant and could not be a major parameter in the experiment. However, spatial water quality such as the most significant changes, for example water temperature, can affect the spatial distribution of fish.

## 3.3. Thermal Stress of Fish

Table 3 and Figure 5 show the results of an estimation of calculated thermal stress from the fish inhabiting the study sites based on the water temperature and FPT (final preference temperature), which showed the largest variation among the water quality factors.

A total of five species of fish were found in the study site. Overall, the proportion of Cyprinidae was high, representing a total of 75.7% of the fish; of these, the bitterling was dominant, accounting for 57.9%. The common freshwater goby showed the lowest proportion at 5.6%. In general, it is a common phenomenon that Cyprinidae has a high distribution in the lentic ecosystems in Republic of Korea. Aforementioned above, physicochemical water quality was not a significant parameter for fish in the study site because it is not a remarkable level of threat to fish. Rather, basic parameters such as water temperature and DO were considered to be more important factors for both distribution and growth of fish due to the relatively large fluctuations.

To compare the thermal stress of different fish species, it is necessary to first identify the tolerance and preference of fish related to heat. Many prior studies that have considered the relationship between heat and fish presented optimal conditions, preferred conditions, spawning conditions, and lethal conditions [33,34,39]. These results provide a type of end point based on the physiological characteristics of fish, and can be utilized as useful data for configuring various standards related to temperature. For example, the optimal temperature is the temperature that maximizes the growth rate of fish, and the FPT refers to the final preference water temperature zone in which the fish can adapt. Finally, the lethal temperature indicates the water temperature zone in which theoretically 50% of the fish may die. These factors are closely related to the exposure time as well as the external water temperature [33].

The crucian carp showed the highest thermal resistance among the fish species in the study site. The FPT of the crucian carp was 26.0–35.8 °C, which was the highest among the species found in the study site, and the FPT used in this study was 30.0 °C. Many preceding physiological studies (on optimal temperature, final preference temperature, lethal temperature, etc.) have found that the crucian carp is a typical warm water fish that prefers warm conditions [40–47].

**Table 3.** Measurement data of relative abundance and characteristics for water temperature parameters of fish fauna in the pond.

| Family | Species | RA [a] (%) | OT [b] (°C) | FPT [c] (°C) | ST [d] (°C) | LT [e] (°C) | FPT of This Study (°C) | References |
|--------|---------|------|------|------|------|------|------|------------|
| Cyprinidae (Acheilognathinae) | Bitterling (*Rhodeus uyekii*) | 57.9 | 12.0–24.3 | 20.2 | 15.0-21.0 | 36.5 | 20.2 | [34,35] |
| Cyprinidae | Crucian carp (*Carassius auratus*) | 17.8 | 25.0–30.0 | 28.0–35.8 | 26.0–28.0 | 38.6–43.6 | 30.0 | [33,40,42–44] |
| Danioninae | Minnow (*Zacco platypus*) | 12.1 | 28.0–30.5 | 29.0 | 26.0 | 32.0 | 29.0 | [48–50] |
| Cobitidae | Chinese muddy loach (*Misgurmus mizolepis*) | 6.5 | 17.8–26.1 | 26.1 | 18.0–26.0 | 30.8 | 26.1 | [51] |
| Gobiidae | Common freshwater goby (*Rhinogobius brunneus*) | 5.6 | 17.0–21.0 | ND [f] | 9.0–15.0 | ND [f] | 23.0 | [52] |

[a] Relative abundance; [b] Optimal temperature; [c] Final preference temperature; [d] Spawn temperature (include development time of egg into larvae); [e] Lethal temperature; [f] ND: no data.

On the other hand, the bitterling showed the lowest thermal resistance among the species. The FPT of the bitterling has been found in many studies to be around 20.0 °C [34,35], and the FPT utilized in this study was 20.2 °C [34]. In general, the bitterling is a cool water fish that prefers a relatively cool environment, unlike the crucian carp [34] and is recognized as a sensitive species as it is vulnerable to water temperature changes.

Figure 5 shows the results of thermal stress over time for the five species of fish found in the study site. Thermal stress increased significantly as the water temperature rises, and showed different patterns depending on the stagnant and circulation conditions and space.

In terms of stagnant and circulation conditions, the thermal stress reduction effect was apparent within the ADP under stagnant conditions. Under stagnant conditions, a total of three fish species (bitterling, Chinese muddy loach, common freshwater goby) in St. 1 was affected by the thermal stress (see Figure 5a). This led to a cumulative increase of thermal stress without a significant recovery period. From highest to lowest thermal stress, the order was bitterling, Chinese muddy loach, and common freshwater goby; the lower the FPT, the higher the thermal stress value. St. 2 showed lower thermal stress than St. 1, resulting in a thermal stress reduction effect of 57.8% for the bitterling and 87.1% for the common freshwater goby (see Figure 5b). In particular, St. 2 can be used as an efficient shelter for water temperature changes, as no thermal stress was found in the Chinese muddy loach of St. 2. In addition, no thermal stress was found in the crucian carp and minnow in both locations of St. 1 and St. 2 under stagnant conditions.

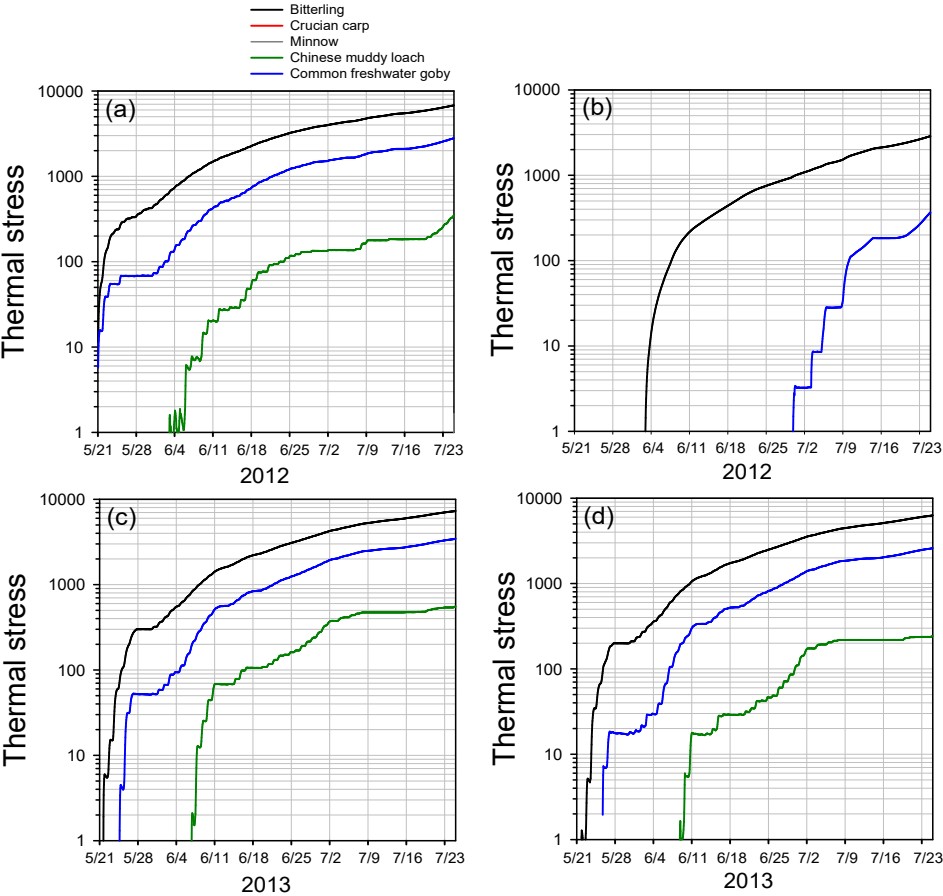

**Figure 5.** Thermal stress results for fish fauna in study site. (**a**) Open space (St. 1) for stagnant conditions in 2012. (**b**) Inside the ADP (St. 2) for stagnant conditions in 2012. (**c**) Open space (St. 1) for circulation conditions in 2013. (**d**) Inside the ADP (St. 2) for circulation conditions in 2013. Crucian carp and minnow showed ND.

Even under the circulation conditions, there was a thermal stress reduction effect within the ADP. As in the stagnant conditions, three fish species (bitterling, Chinese muddy loach, common freshwater goby) showed thermal stress in the circulation conditions, and the overall trend of thermal stress was not much different from the stagnant conditions (see Figure 5c). However, the thermal stress reduction effect of St. 2 was 13.9% for the bitterling, 56.4% for the Chinese muddy loach, and 24.7% for the common freshwater goby, which was lower than that of stagnant conditions, as the shading effect of the ADP was slightly reduced due to internal circulation (see Figure 5d). Nevertheless, compared to the open space of St. 1, the ADP clearly exhibited a thermal stress reduction effect. On the other hand, under circulation conditions, no thermal stress was found in the crucian carp and minnow in both locations of St. 1 and St. 2, as in the stagnant conditions.

The littoral zone generally provides suitable habitat conditions for a variety of fish species. However, due to the high variations in water temperature, both growth and survival of fish were adversely affected [31,45]. Results from this study indicate that the installation of the ADP in the littoral zone can provide stable and optimal water temperature for reproduction or spawning by protecting fish from frequent variations in water temperature.

Fish require higher metabolic rates under high water temperature conditions [32], and continuously high water temperatures accelerate hemoglobin transport within the body of a fish, limiting the oxygen absorption and leading to suffocation or protein modification [46]. Since the experimental results in this study revealed that the water temperature inside the ADP was lower and more stable than that in the open space, the monitored fish were observed to migrate into the ADP when water temperatures rise, and the onsite experiment results proved that considerable fish activities occurred inside the ADP. Thus, the ADP has been phenomenologically proven to be an appropriate space for fish to mitigate the impact of thermal stress on fish.

### 3.4. Diurnal Variations for Thermal Stress of Fish

The two-way ANOVA performed on the water temperature revealed clearly different patterns between St. 1 and St. 2 under stagnant conditions ($p < 0.05$), whereas the inter-site under circulation conditions showed a similar trend between St. 1 and St. 2. As also presented in Figures 6a,b and 7a,b, differences in the values for the water temperature between St. 1 and St. 2 were observed in the diurnal water quality variation. In the case of St. 1, water temperature rapidly increased until 17:00, then gradually decreased, but water temperature in St. 2 showed constant and low standard deviation. In particular, noticeable differences were closely associated with weather conditions, and the diurnal variations in the water quality were greater during the normal season than in the rainy season. These results indicated that both spatial and seasonal influences on the water quality parameters were greater if less agitation of the water body occurred during the normal season. In that sense, St. 1 exhibited greater variations than St. 2, regardless of the weather conditions.

Under stagnant conditions, St. 1 exhibited hourly variations, whereas St. 2 exhibited near constant levels. Especially, the water temperature at St. 2 exhibited no diurnal variations. Although the hourly standard deviation of water temperature was measured to be higher in the rainy season due to the ambient temperature and the movement of the water body induced by the heavy rainfall, the water temperature measured at St. 2 was still more stable than those measured at St. 1 (see Figure 6a,b). This finding confirms that the underground ADP provides a more stable fish shelter than open space, as the water temperature is maintained at constant levels throughout the day, meaning that fish do not experience thermal stress induced by the daily variations in the water temperature.

The results of evaluating the thermal stress based on the average daily cycle of water temperature data are shown in Figures 6 and 7. Through analyzing the daily cycle of thermal stress, it was found that the inside of the ADP more significantly reduces thermal stress under stagnant conditions than in circulation conditions. In terms of the daily cycle trend of thermal stress, the bitterling showed the most rapid increase of thermal stress from 10:00 to 17:00 when the water temperature rises during the normal season (see Figure 6a1), and showed a near-linear increase of thermal stress overall during the rainy season (see Figure 6b1,b2). This phenomenon seems to be related to the diurnal peak size of water temperature and the physiological data (e.g., FPT) of the fish.

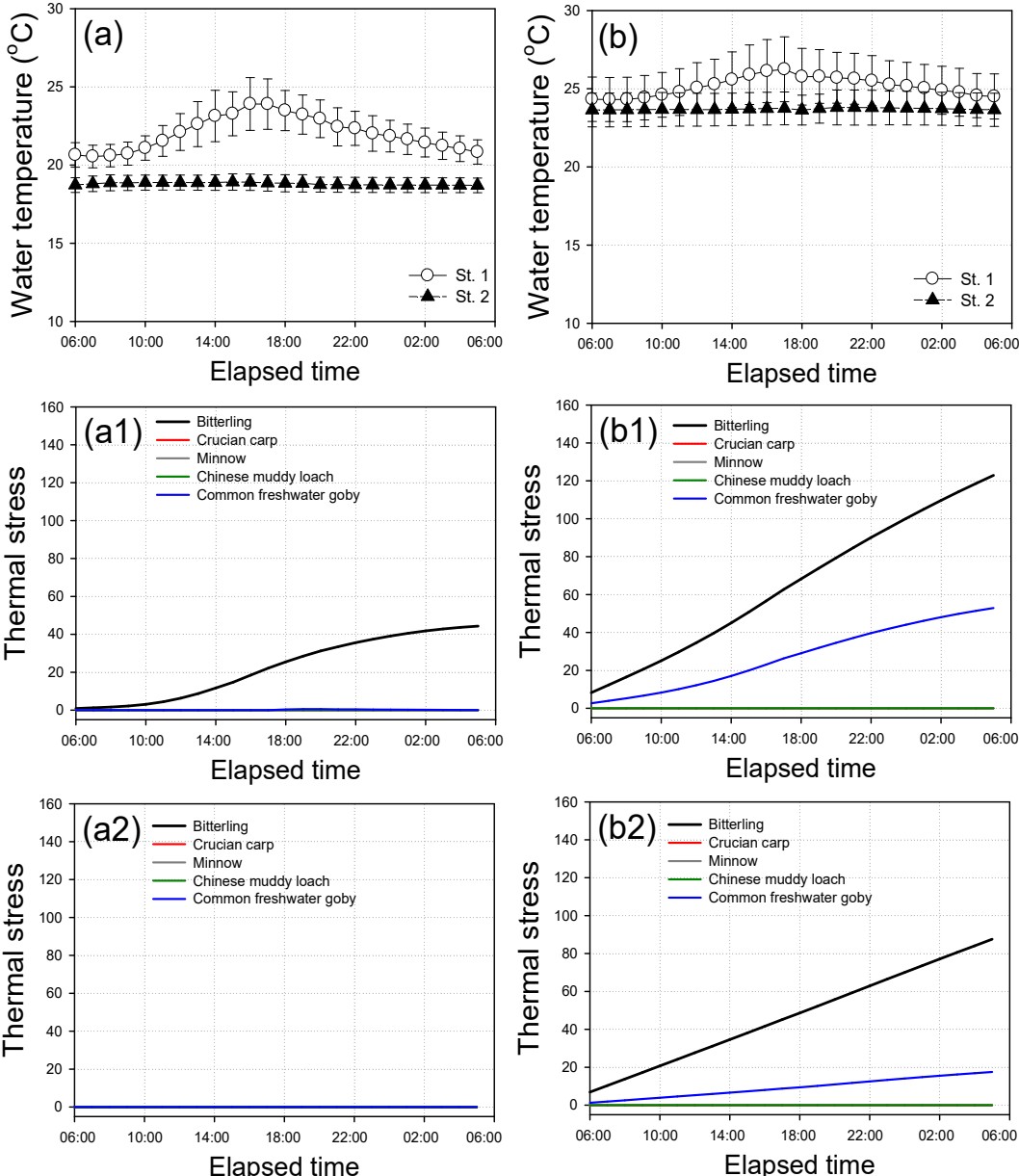

**Figure 6.** Diurnal variations in the average water temperature and thermal stress for the fish fauna inhabiting the study site in 2012. (**a**) Normal season. (**b**) Rainy season. (**a1**) Thermal stress for various fish fauna in St. 1 in the normal season. (**a2**) Thermal stress for various fish fauna in St. 2 in the normal season. (**b1**) Thermal stress for various fish fauna in St. 1 in the rainy season. (**b2**) Thermal stress for various fish fauna in St. 2 in the rainy season. Crucian carp and minnow showed ND.

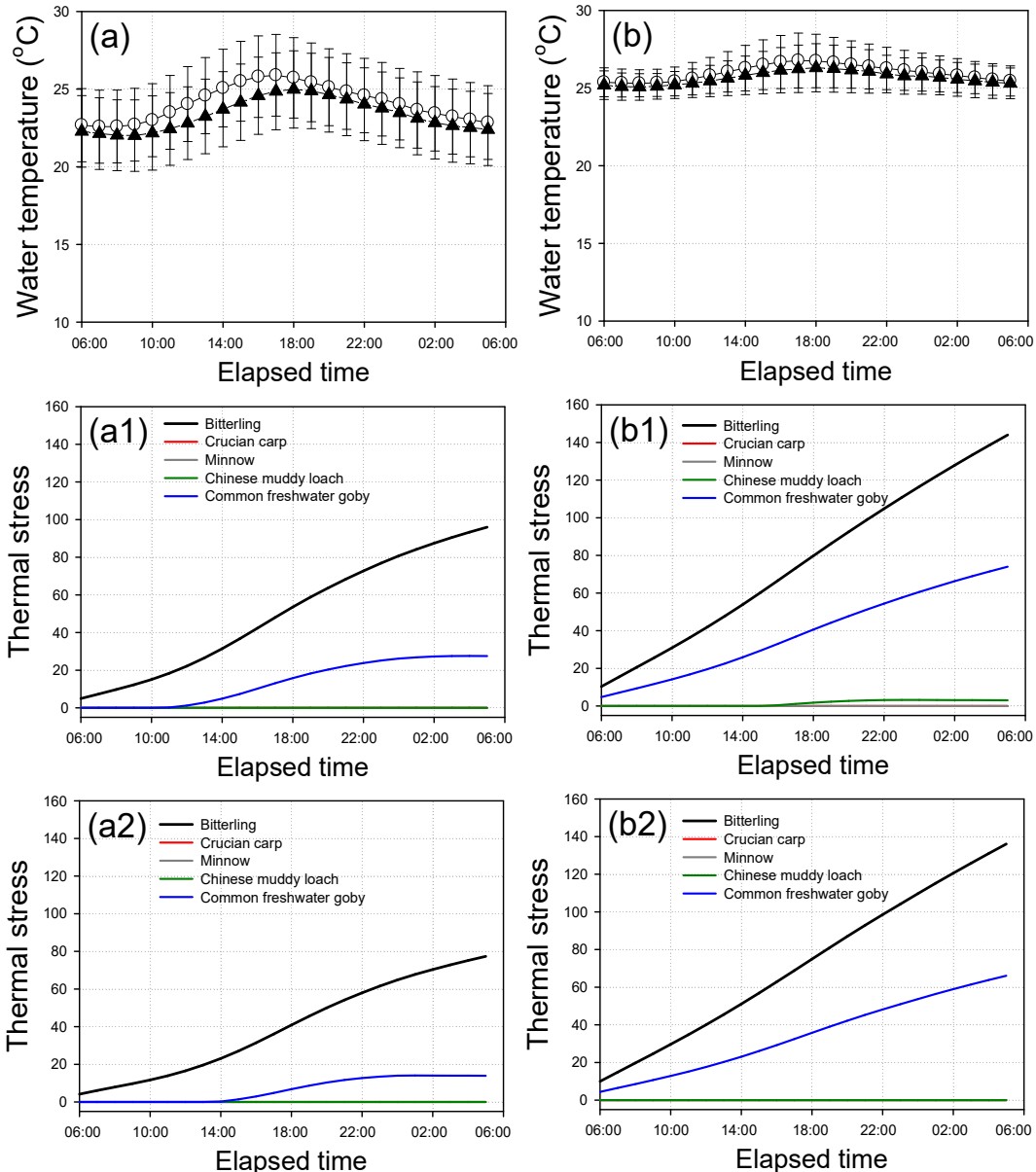

**Figure 7.** Diurnal variations in the average water temperature and thermal stress for the fish fauna inhabiting the study site in 2013. (**a**) Normal season. (**b**) Rainy season. (**a1**) Thermal stress for various fish fauna in St. 1 in the normal season. (**a2**) Thermal stress for various fish fauna in St. 2 in the normal season. (**b1**) Thermal stress for various fish fauna in St. 1 in the rainy season. (**b2**) Thermal stress for various fish fauna in St. 2 in the rainy season. Crucian carp showed ND.

In terms of the efficiency of thermal stress reduction in the ADP, under stagnant conditions, the thermal stress of the bitterling and common freshwater goby was reduced by 22.8–100.0% and by 66.8–100.0%, respectively; under the circulation conditions and reduced by 5.5–19.4% and by 10.7–49.4%, respectively. In particular, the thermal stress of fish showed higher efficiency under conditions when the water body was stagnant or during normal season. These results are attributed to the enhancement of the bitterling survival in the study site. This is due to the water temperature of the study site being below the lethal temperature, as the bitterling showed the highest thermal stress of all fish under a physicochemical environment, but reducing the valid part of thermal stress gave it

the dominant position in the survival competition against other species. In addition, ADP can create the preference environment (e.g., water column) [19] and appropriate winter habitat for positive conditions to survival of the bittering, and further study is in progress to validate the ecological functions of the ADP.

Through this analysis, we were able not only to identify when the thermal stress intensively increases, but also to estimate the trend and efficiency of the diurnal of thermal stress on the fish species in the study site. However, because the data were simulated as the mean values of the water temperature, real-time water temperature fluctuations were not fully applied, so the thermal stress of some species may not have been reflected, or may have a lower value. Nevertheless, the diurnal analysis of water temperature and thermal stress are meaningful since the trend of values can be identified over time. Further study is required to investigate the thermal stress in consideration of fish size and life stage based on this study.

Also, in this respect, it can be assumed that the water temperature and fluctuations are important factors causing thermal stress to fish. The diurnal temperature variations in fresh water during summer were reported to sometimes exceed 10 °C depending on the region [10]; in another study, a rise in water temperature of 1 °C was reported to reduce habitat by 31% on average [47]. Therefore, the underground ADP may reduce heat-related mortality, cortisol levels, and disease for fish in a period of harsh heat.

## 4. Conclusions

To evaluate the function of a fish shelter in mitigating the impact of accumulated stress, the water quality inside the ADP was investigated for two consecutive years during the early summer period when thermal stress tends to rise in fish. The water temperature, pH, DO, and EC were measured on an hourly basis in the open water (i.e., St. 1) and inside the ADP (i.e., St. 2), and a phenomenological study was performed by dividing the seasons into normal and rainy and the environments into stagnant and circulating environments.

The key findings observed in this study can be summarized as follows. First, the water temperature inside the ADP is lower than outside it by 1.7–3.7 °C in stagnant conditions, and by 0.6–0.7 °C in circulation conditions on average during early summer. Secondly, the water temperature inside the ADP exhibits lower fluctuations and diurnal variations compared with the open water space. In addition, diurnal water temperature fluctuations inside the ADP were steady, but in the open space showed ≥4 °C. Thirdly, thermal stratification occurs but is temporarily disturbed due to the mixing from the forced circulation and the rainwater input through rainfall events. Finally, ADP was able to significantly reduce the thermal stress of the fish in the study site during early summer, and as a result, it was considered as the shelter for bitterling, a cool water fish species, to survive during the hot summer.

Through the experiment, the ADP was found to provide constant and optimal water temperature for living and spawning for bitterling, which dominated in the experimental pond during early summer. Also, the ADP provides a more stable fish shelter than open water since the water temperature is maintained at a constant level, thus enabling the fish to avoid the thermal stress induced by diurnal variations in the water quality. Moreover, the ADP can create various environments depending on the depth, and provide various aquaculture environments suitable for different fish species. In particular, as a result of this study, ADP can be applied to shallow water spaces such as garden ponds, park ponds, and other artificial wetlands because it exhibits effective thermal stress relief of fish in the small aquatic ecosystem. Therefore, the deployment of the ADP appears to offer a practical alternative for effective fishery resources management to improve the species diversity of fish communities in an artificial freshwater ecosystem.

**Author Contributions:** Formal analysis, C.H.A., S.L. and J.C.J.; Methodology, C.H.A.; Supervision, H.M.S. and J.R.P.; Writing—original draft, C.H.A.; Writing—review & editing, C.H.A., S.L. and J.C.J.

**Funding:** The research was funded by MOLIT (Ministry of Land, Infrastructure and Transport) and ME (Ministry of Environment), Republic of Korea.

**Acknowledgments:** This subject was supported by "Recycling plan for the ecological pond of rainwater reservoir by LID" (code 18TBIP-C124749-02) and "Development of algae management using stream structures in the stream" (code 18AWMP-B112149-04) funded by MOLIT (Ministry of Land, Infrastructure and Transport) and ME (Ministry of Environment), Republic of Korea.

**Conflicts of Interest:** The authors declare no conflict of interest.

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
