# Peer review of "Assessment of Water Quality and Thermal Stress for an Artificial Fish Shelter in an Urban Small Pond during Early Summer"

_water, doi:10.3390/w11010139_

Round 1
Reviewer 1 Report
Manuscript ID water-407018 - Review
General comments:
I became very interested in this paper after reading its abstract and had been looking forward to read, learn, and offer a review. I found d this paper well written, to the point, with sufficient introduction, and well laid out. I only have a few comments to offer; however, few are critical enough that will require a closer look at some of the analysis. Otherwise, a good paper and good work.
Specific comments (numbers reflect line numbers):
62 “That can affect......pond design.” I cannot understand this sentence. Consider rewriting it.
72-76. It indicates that holes were perforated at four sides of the ADP. So was the ADP sitting on the bottom or was the top of it leveled with the bottom as shown in Fig 1? And if the ADP was sunk in, as shown, then why would there be a need for holes on the sides?
--Also. It indicates that the pump worked every 30 min for 30 min each time (30 min. intervals). Correct? So it replaced the total volume of the ADP in about 2.2 hr? Correct? If so how were these values derived?
95 So sensors were installed at each of these two depths. Correct?
96 Were the data collection synchronized with the pump action? And how?
131 An open parentheses is missing.
192 Fig 3.
o Show the depth the data was collected at.
o Fonts are too small. Increase all sizes.
o Reduce the number of increments on the right hand side axis to allow you to increase the fonts.
203-207. So the stratification (depth dependent) was formed in the 30 min that pump was off? Sorry if I missed it. But this is critical to the paper.
218 Fonts too small. Increase the size. For the X axis. Reduce the number of the increments (increase the values in between, so marks are every 4 degree for Temp. Vs 1 degree). So you can increase the font size.
263 I am not a fish expert. But I cannot follow how your work here can lead to your conclusion that one of the fish can become dominant (line 408). Given that habitat is continuously shifting from mixing to non-mixing environment.
Author Response
Point 1: (Line 62) “That can affect......pond design.” I cannot understand this sentence. Consider rewriting it.
Response 1: The authors sincerely appreciate the reviewer's helpful and insightful comments. Also, the authors fully understand why the reviewer concerned about the sentence for this manuscript. Thus, the authors have changed the sentence to guide this study (see lines 81-82).
To evaluate the function of a fish shelter in mitigating the impact of accumulated thermal stress, the water quality inside the ADP was investigated for two years during the period from early summer when thermal stress tends to rise in fish. The specific objectives of this study were (1) to confirm the characteristics of the water quality inside the ADP through a phenomenological approach, (2) to evaluate utilization possibilities for the ADP buried at the bottom of a shallow pond, and (3) to perform a quantitative evaluation of the thermal stress that can affect the ecological health for fish.
Point 2: (Line 72-76) It indicates that holes were perforated at four sides of the ADP. So was the ADP sitting on the bottom or was the top of it leveled with the bottom as shown in Fig 1? And if the ADP was sunk in, as shown, then why would there be a need for holes on the sides? Also. It indicates that the pump worked every 30 min for 30 min each time (30 min. intervals). Correct? So it replaced the total volume of the ADP in about 2.2 hr? Correct? If so how were these values derived?
Response 2: The authors sincerely appreciate the reviewer's helpful and insightful comments. As the reviewer said, the top of the ADP leveled with the bottom as shown in Figure 1. Considering the horizontal movement as well as vertical movement of the fish, the holes in its cover and the spaces for its sides help the fish move. In addition, the holes in its cover and the spaces for its sides can help water circulation between open space and inside the ADP. Especially when the underwater pump (30 min intervals) was operated (in 2013), the water circulated between open space and inside the ADP, and the HRT of the ADP is about 2.25 hr. Thus, the authors have modified the explanations to make clear of the detail specifications of the ADP (see lines 112-114).
Figure 1. (a) Conceptual view of the study site. The y-axis is the water level and the x-axis is the bottom materials (not to scale). The intake point of the underwater pump is at a water depth of 1.0 m inside the ADP. The water quality monitoring sensors are at a water depth of 0.4 m in open space (St. 1) and 1.5 m inside the ADP (St. 2). (b) Pictorial view of constructed ADP in the bottom of the pond. The main body (fish shelter) is buried underground and fish can vertically or horizontally move through the holes (fish passages) in the cover and sides.
Point 3: (Line 95) So sensors were installed at each of these two depths. Correct?
Response 3: Thank you for the detailed comments of reviewers. Yes, the water quality monitoring sensors were installed each of two points (0.4 m depth in open space of pond and 1.0 m depth inside the ADP), respectively, to compare water quality changes in and out of the ADP. Thus, the authors have modified the sentence (see line 127).
Two water quality measurement points were designated at St. 1 for the open space and at St. 2 inside the ADP (see Figures 1 and 2). The water quality monitoring sensors (XLM6000, YSI, USA) were installed at depths of 0.4 m (representing the open space) and 1.0 m (representing the inside the ADP) for St. 1 and St. 2, respectively, to measure the water temperature, pH, DO, and EC.
Point 4: (Line 96) Were the data collection synchronized with the pump action? And how?
Response 4: The authors sincerely appreciate the reviewer's helpful comments. Every hour based data (e.g. water temperature, pH, DO, and EC) were measured by remotely data-storable water quality measurement. Therefore, we retrieved the sensor once a month and confirmed the data. And depth-dependent data inside of ADP was used portable devices (550A, YSI, USA; 63, YSI, USA) during non-pump action period or with the underwater pump off for a while. Thus, the authors have cleared the methodology (see line 129, lines 132-133).
Measurements were conducted remotely every hour for 132 days from May 21 to July 25 in 2012 and 2013. Since the monsoon season in the Republic of Korea generally begins in June and lasts for approximately one month [19], the measured data was classified into normal and rainy season. The depth-dependent data inside the ADP was generated by portable devices (550A, YSI, USA; 63, YSI, USA) during non-pump action period or with the underwater pump off for a while.
Point 5: (Line 131) An open parentheses is missing.
Response 5: Thank you for the detailed comments of reviewers. We have supplemented as follows this sentence. Thus, the authors have rewrite the manuscript (see line 167).
Thermal stress (t) = Thermal stress (t - dt) + (Accumulation - recovery)dt (1)
Point 6: (Line 192) Fig 3. Show the depth the data was collected at. Fonts are too small. Increase all sizes. Reduce the number of increments on the right hand side axis to allow you to increase the fonts.
Response 6: Thank you for the detailed comments of reviewers. We have changed the graph as follows. And the number of increments was reduced on the right hand side axis to increase the font size. Thus, the authors have modified the manuscript (see lines 228-233).
Figure 3. Variations in the water temperature, pH, DO, and EC values of St. 1 and St. 2 during the normal and rainy seasons under stagnant (a, b, c, d; 2012) and circulation (e, f, g, h; 2013) conditions. The water depth kept constant, regardless of time order, and the maximum water depth showed 0.69±0.01 m in 2012 and 0.69±0.03 m in 2013 respectively in this period (n=7).
Point 7: (Line 203-207) So the stratification (depth dependent) was formed in the 30 min that pump was off? Sorry if I missed it. But this is critical to the paper.
Response 7: The authors sincerely appreciate the reviewer's helpful and insightful comments.
Stratification occurred mainly in the stagnant condition, but not in the circulation condition (see Figures 4).Thus, the authors have corrected the manuscript (see line 245).
The fact that there was no variation according to the water depth indicated that the circulation ensured relatively homogenous mixing of the water body inside the ADP due to the current generated by the underwater pump.
Point 8: (Line 218) Fonts too small. Increase the size. For the X axis. Reduce the number of the increments (increase the values in between, so marks are every 4 degree for Temp. Vs 1 degree). So you can increase the font size.
Response 8: Thank you for the detailed comments. We have changed the graph as follows. And the number of increments was reduced on the right hand side axis for increasing the font size. Thus, the authors have changed the manuscript (see line 259).
Point 9: (Line 408) I am not a fish expert. But I cannot follow how your work here can lead to your conclusion that one of the fish can become dominant. Given that habitat is continuously shifting from mixing to non-mixing environment.
Response 9: The authors sincerely appreciate the reviewer's helpful and insightful comments. Also, the authors fully understand why the reviewer concerned about the sentence for this manuscript. In this study, one of the important functions of ADP is to secure shelter that can be avoided when water temperature exceeds the physiological limits of fish. This can be more effective for cool water species such as bitterling which showed the highest thermal stress in this study. Another function of the ADP is to ensure a sufficient water column. Bittering, water column species, prefer to live in the water layer, and if sufficient water column is secured in the shallow pond like the study site, it will be positive for their survival. Though not covered in this study, the presence of deep water in the shallow pond is considered to be an important point for habitat in winter when the water volume is reduced or the surface is iced. Thus the presence of ADP will be considered to contribute to a variety of factors for dominant of bittering. Especially, it is considered that the thermal stress reduction which is the result of this study is one of them. Thus, the authors have modified the manuscript (see lines 422-423, lines 426-428, lines 458-459).
These results are attributed to the enhancement of the bitterling survival in the study site. This is due to the fact that the water temperature of the study site was below the lethal temperature, and as the bitterling showed the highest thermal stress of all fish under a physicochemical environment, but, reducing valid part of thermal stress gave it the dominant position in the survival competition against other species. In addition, ADP can create the preference environment (e.g. water column) [19] and appropriate winter habitat for positive conditions to survival of the bittering, and further study is in progress to validate the ecological functions of the ADP.
Finally, ADP was able to significantly reduce the thermal stress of the fish in the study site, and as a result, it was considered as the shelter for bitterling, a cool water fish species, to survive successfully during the hot summer season.

Reviewer 2 Report
This manuscript presents the long-term change of the water quality of pond with artificial fish shelter. This study is important for urban ecosystem management. I do see the need for some clarifications, however, and I hope that you can share my arguments below.
1. Chronological change of water level is thought to be strongly affected the temporal change in water quality. If the data of water level is available, it should be described in Figure 3.
2. The investigation site of fish fauna is not clear. If the fish fauna survey was conducted in the ADP and other areas in the pond, discussion of the relationship between water quality and fish fauna should be added.
3. Thermal stress is assumed to vary regionally and individually. The rationale of the compatibility of the previous studies should be described. If you have the data of the seasonal change of the fish fauna, a relationship between predicted thermal stress and the seasonal change of fish fauna should be discussed.
Author Response
Point 1: Chronological change of water level is thought to be strongly affected the temporal change in water quality. If the data of water level is available, it should be described in Figure 3.
Response 1: The authors sincerely appreciate the reviewer's helpful comments. In this study, the maintenance water utilized collected rainwater and tap water. This pond is basically lentic system, but when the water level drops, the water pump supplies the necessary amount of flow. In addition, when the large amount of flow is supplied by rainfall, the exceeded amount of flow is discharged to the outlet. Therefore, even though the pond is small in size, it can always have a constant level of water volume and depth. Thus, the authors have changed the manuscript (see lines 91-95, lines 231-233).
The mesocosm experiments were conducted in a pond at the Korea Institute of Civil Engineering and Building Technology (KICT) in Gyeonggi-do, Republic of Korea. As shown in Figure 1, the specifications of the pond (the study site) are as follows: surface area (110 m2), average water depth (about 0.5 m), and maximum water depth (about 0.7 m). Both gravel (diameter ≤ 60 mm) and sand (diameter ≤ 2 mm) were used as bed materials, and bentonite liner (5 cm) was used as an impermeable layer. The water level was maintained as constant throughout the year using both collected rainwater and tap water, with a water level sensor and underwater pump installed at the water inflow point. This pond is basically lentic system, but when the water level drops, the underwater pump supplies the necessary amount of flow. In addition, when the large amount of flow is supplied by rainfall, the exceeded amount of flow is discharged to the outlet. Therefore, even though the pond is small in size, it can always have a constant level of water volume and depth.
Figure 3. Variations in the water temperature, pH, DO, and EC values of St. 1 and St. 2 during the normal and rainy seasons under stagnant (a, b, c, d; 2012) and circulation (e, f, g, h; 2013) conditions. The water depth kept constant regardless of time order, and the maximum water depth showed 0.69±0.01 m in 2012 and 0.69±0.03 m in 2013 respectively in this period (n=7).
Point 2: The investigation site of fish fauna is not clear. If the fish fauna survey was conducted in the ADP and other areas in the pond, discussion of the relationship between water quality and fish fauna should be added.
Response 2: The authors sincerely appreciate the reviewer's helpful and insightful comments. The investigation site was the entire pond and the blocking net (mesh 3 × 3 mm) was placed on the cover of the ADP prior to fish monitoring to minimize the interference of the fish shelter. In the field investigation, the ADP was limited to quantitative fish monitoring inside ADP due to its small entrance. Also, in this study, fish monitoring part merely aimed to understand the overall distribution of fish in ponds, so fish monitoring was mainly conducted in the open space except fish shelters. Lastly, we have changed this sentence for discussion of the relationship between water quality and fish fauna. Thus, the authors have added the sentence (see lines 150-152, lines 311-316).
The fish were monitored four times over the study period using capture per unit effort (CPUE) method by kick net (mesh 3 × 3 mm) in the study site. The investigation site was the entire pond and the blocking net (mesh 3 × 3 mm) was placed on the cover of the ADP prior to fish monitoring to minimize the interference of the fish shelter. Relative abundance result was calculated on average based on the captured fish population.
A total of five species of fish were found in the study site. Overall, the proportion of Cyprinidae was high, representing a total of 75.7% of the fish; of these, the bitterling was dominant, accounting for 57.9%. The common freshwater goby showed the lowest proportion at 5.6%. In general, it is a common phenomenon that Cyprinidae has a high distribution in the lentic ecosystems in Republic of Korea. Aforementioned above, physicochemical water quality was not a significant parameter for fish in the study site because it is not a remarkable level of threat to fish. Rather, basic parameters such as water temperature and DO were considered to be more important factors for both distribution and growth of fish due to the relatively large fluctuations.
Point 3: Thermal stress is assumed to vary regionally and individually. The rationale of the compatibility of the previous studies should be described. If you have the data of the seasonal change of the fish fauna, a relationship between predicted thermal stress and the seasonal change of fish fauna should be discussed.
Response 3: The authors sincerely appreciate the reviewer's helpful and insightful comments. Also, the authors fully understand why the reviewer concerned about the sentence for this manuscript. In this study, it is an important assumption that thermal stress is closely related to the fluctuation of water temperature. Also, water temperature in natural state can show diurnal variations of 4 to 10 °C, and especially rapid changes of water temperature can give physiological stimulation to sensitive fish (Bevelhimer and Bennett, 2000). In addition, according to previous researches, the thermal stress of fish repeated acclimation and recovery according to the rise and fall of water temperature (Bevelhimer and Bennett, 2000), so fish will instinctively utilize safe space if there is a fish shelter. The risk of thermal stress has been also suggested in the previous study of Carveth et al. (2007), LT50 (50% survival rate per 30 min at various water temperatures) of spikedace was 32.1 °C and the mortality rate increased sharply at water temperature above that. In this respect, the rationale of the compatibility of the previous studies is considered to be related to the content of the thermal stresses covered in this study.
In the natural state, the fish instinctively use the pool to find the appropriate water temperature range (Harvey and Berg, 1991; Matthews et al., 1994), but artificial water space is monotonous and difficult to secure structural diversity for fish shelter. Therefore, it is considered that the introduction of shelter such as ADP of this study is important in securing diversity of ecological structure. In conclusion, in the study, we estimated the thermal stress of fish in the urban pond and discussed the application of the ADP. However, since the characteristics of fish are different from each other, details such as change of the fish fauna and physiology of fish should be dealt with in future studies. Thus, the authors have modified the manuscript (see lines 39-40, lines 44-64, line 435-436).
In particular, fish are very sensitive to water temperature [6,8]. Excessively high water temperatures or high diurnal variations cause thermal stress in fish, and this can be aggravated through accumulation [9]. As such, fish are easily exposed to continuous cycles of stress and de-stress depending on the rise and fall of the water temperature [10].
In this respect, summer, the season of rising temperatures, sometimes can be a harsh period for fish. For example, an excessive increase in temperature can induce numerous physiological changes in the body of fish. Previous studies have shown that fish exposed to high temperatures have an adverse effect on protein damage, hormonal changes, and high mortality due to thermal stress [12,13]. Therefore, adequate water management focused on water temperature is closely related to the health of the fish, so the appropriate preparation is required especially in the season when water temperature is rising.
As mentioned, since excessive rises in water temperature in aquatic environments can be detrimental to the reproduction and growth of fish, there has been ongoing research into the functions of fish shelters [14-19,23,24]. Previous studies have found that fish shelters can be used to prevent untimely predator encounters [14,15], to enable the survival of physical disturbance events such as floods or droughts, and to maintain or increase fish populations through enhancing the survival rates [19]. However, most studies have emphasized the complexities of fish shelter structures, and have focused on the vulnerability of prey fish. For this reason, a phenomenological approach is needed to evaluate the function of a fish shelter in mitigating the impact of accumulated thermal stress.
Recently, a large number of water spaces have been introduced into urban areas and gardens to improve ecological functions [20,21]. However, the problems of these facilities are limited to habitat of fish fauna because they are easily exposed to the thermal stress due to its insufficient water volume and shallow water depth. In these circumstances, an additional method of securing depth, such as the deep pool, can be used as an alternative. Deep pool in their natural state is generally known to mitigate the elevated water temperatures due to their geometric structure and depth, which can contribute to increased survival rate and efficient habitat of fish [22,23].
Through this analysis, we were able not only to identify when the thermal stress intensively increases, but also to estimate the trend and efficiency of the diurnal of thermal stress on the fish species in the study site. However, because the data were simulated as the mean values of the water temperature, real-time water temperature fluctuations were not fully applied, so the thermal stress of some species may not have been reflected, or may have a lower value. Nevertheless, the diurnal analysis of water temperature and thermal stress are meaningful since the trend of values can be identified over time. Further study is required to investigate the thermal stress in consideration of fish size and life stage based on this study.

Reviewer 3 Report
The presented manuscript "Assessment of Water Quality and Thermal Stress for an Artificial Fish Shelter in an Urban Small Pond during Early Summer" is adressing an interesting topic, but in the current form missing a discussion of possible applications of the findings.
First of all the manuscript lacks a proper description of the ADP. Please explain the Setting and why it was build; not only refer to a previous study
If you have a bentonite layer, how thick is the Substrate layer on top? You are working on a very small shallow pond, can you provide Information regarding inflow / outflow respectively Evaporation?
You mention Stagnation and circulation (line 77 // line 127 // line 158 ff. // Fig. 04 !!), I suspect those processes to appear in such a shallow garden pond! And your values (Fig. 04) do not reflect what you describe!
You mention that BP was measured through applying 760 mmHg --> at what height (a.s.l.) the pond is situated?
Provide a reference for the Van Dorn sampler (line 110).
The rainfall data comes from a Station nearby - add details (distance, ...) - line 114
Do you mean a shelter for all spp. or only Bitterling (line 178)?
Move Table 1 + 2 into the Annex.
"Physical water Quality was slightly higher..." - according to what?
Check spelling in line 245.
In Figure 5 only 3 species are included, but 5 spp. are mentioned above. maybe you add the temperature to the Graphs also?
Figure 6 / 7: (a) + (b) Change scale --> e.g. 15 to 30
again only 3 species are included, but 5 spp. are mentioned in each graph. maybe you add the temperature to the Graphs also?
What about different size classes / life stages (line 383-386)?
Where do you suggest to apply a ADP? in a garden pond, in hatcheries --> possible application of your findings need to be discussed (line 391-392 + 411-416)!
Try to attract the reader. You should also highlight and discuss the relevance! A detailed discussion on the performance, including a comparison of your designs vs. others in the literature and how the findings can be applied to general scientific community. And finally try to adress how the findings can be used in e.g. in a hatchery.
Author Response
Point 1: The presented manuscript "Assessment of Water Quality and Thermal Stress for an Artificial Fish Shelter in an Urban Small Pond during Early Summer" is addressing an interesting topic, but in the current form missing a discussion of possible applications of the findings.
Response 1: The authors sincerely appreciate the reviewer's helpful and insightful comments. Also, the authors fully understand why the reviewer concerned about the sentence for this manuscript. The pond located in the urban area are widely used in the ecological aspect, but it is easily exposed to the thermal stress due to shallow water depth and shortage of fish shelter. Therefore, if a fish shelter such as the ADP is applied to shallow or small aquatic ecosystem (garden or park pond, artificial wetland etc.), it will be effective in managing the thermal stress of the fish. In addition, as a result of this study, relative constant water temperature is maintained inside the ADP, so that the spawning period of a specific fish can be sustained. In this respect, the installation of the fish shelters will improve the survival rate of fish. Thus, the authors have modified the manuscript (see lines 465-468).
Through the experiment, the ADP was found to provide constant and optimal water temperature for living and spawning for bitterling that dominated in the experimental pond during early summer. Also, the ADP provides a more stable fish shelter than open space since the water temperature is maintained at constant levels, thus enabling the fish to avoid the thermal stress induced by the diurnal variations in the water quality. As well as, the ADP can create various environments depending on the depth, and provide various aquaculture environments suitable for different fish species. In particular, as a result of this study, ADP can be applied to shallow water spaces such as garden pond, park pond, and other artificial wetlands because ADP exhibits effective thermal stress relief of fish in the small aquatic ecosystem. Therefore, the deployment of the ADP appears to offer a practical alternative for effective fishery resources management to improve species diversity of fish communities in a freshwater ecosystem.
Point 2: First of all, the manuscript lacks a proper description of the ADP. Please explain the Setting and why it was build; not only refer to a previous study.
Response 2: The authors sincerely appreciate the reviewer's helpful and insightful comments. Also, the authors fully understand why the reviewer concerned about the sentence for this manuscript. We have newly supplemented this sentence. Thus, the authors have modified the manuscript (see lines 65-76).
The ADP is developed with the aim of providing shelter to improve fish survival and to mitigate thermal stress in shallow ponds reflecting the above considerations. The ADP is derived from traditional pools that was used in the paddy field in Republic of Korea, and has many advantages in terms of the ecological habitat for freshwater organisms. The basic structure of the ADP is a cuboid consisting of a cover and a main body; the main body is the space of the fish shelter, and the cover consists of holes (diameter 0.2 m) and lateral entrances (height 0.2 m) allowing fish to move. The cover is not only a pathway for fish to evacuate into the ADP, but also a structures that do not fall into deep space when people manage ponds. In the previous study, the ADP is an underground structure that secures a shelter space for fish to escape from adverse conditions during the dry season [19]. Also, ADP was verified to be both an effective fish shelter and a natural habitat area for endangered fish species [19]. Therefore, in this study, we tried to investigate water quality characteristics and to verify the ability to mitigate thermal stress of fish when ADP was applied to a shallow pond.
Point 3: First if you have a bentonite layer, how thick is the Substrate layer on top? You are working on a very small shallow pond, can you provide Information regarding inflow / outflow respectively Evaporation?
Response 3: The authors sincerely appreciate the reviewer's helpful comments. We initially set the bentonite thickness to 5 cm. The relevant part is reflected as below. In this study, the maintenance water utilized both harvested rainwater and tap water. This pond is basically lentic system, but when the water level drops, the water pump with level sensor supplies the necessary amount of flow. In addition, when the large amount of flow is supplied by rainfall, the exceeded amount of flow is discharged to the outlet. Therefore, even though the pond is small in size, it can always have a constant level of water volume and depth. Thus, the authors have changed the manuscript (see line 89, lines 91-95, lines 231-233).
The mesocosm experiments were conducted in a pond at the Korea Institute of Civil Engineering and Building Technology (KICT) in Gyeonggi-do, Republic of Korea. As shown in Figure 1, the specifications of the pond (the study site) are as follows: surface area (110 m2), average water depth (about 0.5 m), and maximum water depth (about 0.7 m). Both gravel (diameter ≤ 60 mm) and sand (diameter ≤ 2 mm) were used as bed materials, and bentonite liner (5 cm) was used as an impermeable layer. The water level was maintained as constant throughout the year using both harvested rainwater and tap water, with a water level sensor and underwater pump installed at the water inflow point. This pond is basically lentic system, but when the water level drops, the underwater pump supplies the necessary amount of flow. In addition, when the large amount of flow is supplied by rainfall, the exceeded amount of flow is discharged to the outlet. Therefore, even though the pond is small in size, it can always have a constant level of water volume and depth.
Figure 3. Variations in the water temperature, pH, DO, and EC values of St. 1 and St. 2 during the normal and rainy seasons under stagnant (a, b, c, d; 2012) and circulation (e, f, g, h; 2013) conditions. The water depth kept constant, regardless of time order, and the maximum water depth showed 0.69±0.01 m in 2012 and 0.69±0.03 m in 2013 respectively in this period (n=7).
Point 4: You mention Stagnation and circulation (line 77 // line 127 // line 158 ff. // Fig. 04 !!), I suspect those processes to appear in such a shallow garden pond! And your values (Fig. 04) do not reflect what you describe!
Response 4: The authors sincerely appreciate the reviewer's helpful and insightful comments. Our study site was conducted in an urban shallow pond. As you know, this type of ponds are almost completely mixed because their depths are shallow (see below the figure (a)). However, after the ADP was installed, some water quality parameters (water temperature, DO etc.) showed the stratification effects inside the ADP (stagnant condition) during 2012 (see below the figure (b)). This phenomenon is probably due to the fact that the circulation between inside the ADP and open space was not significant because the ADP was installed on the bottom of the pond and the inside the ADP was stagnant. To solve this problem, we installed an underwater pump inside the ADP to induce the artificial circulation (circulation condition) during 2013 (see below the figure (c)). In other words, the stagnant and circulation conditions in this study are based on the ADP water flow. In conclusion, this study evaluated the thermal stress of fish on stagnant and circulation conditions under installed ADP. Thus, the authors have corrected the manuscript (see lines 103-104, lines 195-197, line 261).
To evaluate the effects of both stagnation and circulation of pond water on the water quality inside and outside the ADP, the underwater pump was not operated in 2012, but was operated in 2013. These attempts can compare and assess the thermal stress of fish according to the application type of the ADP.
Different trends in water quality parameters within the ADP were observed under the stagnant [see Figure 3(a)–3(d)] and the circulation conditions [see Figures 3(e)–3(h)]. In other words, the stagnant and circulation conditions in this study are based on the ADP flow conditions by applying the underwater pump.
Figure 4. Vertical variations in the water temperature, pH, DO, and EC values inside the ADP under the stagnant (a, b, c, d; 2012) and the circulation (e, f, g, h; 2013) conditions by the underwater pump (n=10). The red dots represent the point at 0.1 m depth in water of the open space.
Point 5: You mention that BP was measured through applying 760 mmHg --> at what height (a.s.l.) the pond is situated?
Response 5: Thank you for the detailed comments of reviewers. The height of this study area is about 10 m above sea level, and we measured the DO saturation by assuming 1 atm as 760 mmHg. Thus, the authors have corrected the manuscript (see line 135).
The barometric pressure (BP) was assumed to be 760 mmHg (i.e., 1 atm),
Point 6: (Line 110) Provide a reference for the Van Dorn sampler.
Response 6: Thank you for the detailed comments of reviewers. Thus, the authors have added the reference (see line 143, lines 537-538).
Samples for physicochemical water quality were collected using a Van Dorn sampler [27].
27. Howmiller, R.P.; Sloey, W.E., A Horizontal water sampler for investigation of stratified waters. Limnol. Oceanog. 1969, 14, 2, 291–292.
Point 7: (Line 114) The rainfall data comes from a Station nearby - add details (distance)
Response 7: Thank you for the detailed comments of reviewers. Thus, the authors have added the information (see lines 147-148).
Rainfall data were obtained from the database from the Gimpo Airport of the Korea Aviation Meteorological Agency (KAMA) about 12 km away from the study site.
Point 8: (Line 178) Do you mean a shelter for all spp. or only Bitterling?
Response 8: The authors sincerely appreciate the reviewer's helpful comments. Shelter mentioned in this manuscript is aimed at all fish in the pond. The basic function of the ADP is not only to reduce thermal stress as in this study, but also to secure a minimum survival space for fish in shallow ponds. However, from the perspective of thermal stress, it is possible to target fish that are vulnerable to heat, including bittering. Thus, the authors have changed the manuscript (see line 215).
Since the ADP was proved to provide constant and optimal water temperature for living and spawning (i.e., 15–21 °C) for the bitterling that dominated the experimental pond [33-35], the ADP can serve as a fish shelter for the fish vulnerable to heat during the periods of high water temperatures.
Point 9: Move Table 1 + 2 into the Annex.
Response 9: The authors sincerely appreciate the reviewer's helpful comments. However, since Table 1 and 2 deals with the “3. Results and Discussions (3.1. Water temperature, pH, DO, EC; 3.2. Physicochemical water quality) of the manuscript respectively, it is rational to be within the main text rather than the annex. Thank you.
Point 10: "Physical water Quality was slightly higher..." - according to what?
Response 10: The authors sincerely appreciate the reviewer's helpful comments. In this study, physicochemical water quality was measured in 2012 and 2013. Although all water quality parameters in 2013 are slightly higher than in 2012, it is not a major factor in this study because there was no significant difference and not a remarkable level of threat to fish. Thus, the authors have corrected the manuscript (see lines 284-285).
The results from the physicochemical water quality analyses performed in 2012 and 2013 were summarized in Table 1 and Table 2, respectively. Overall, the physicochemical water quality in 2013 was slightly higher than in 2012 in terms of seasonality, but there was no significant difference between 2012 and 2013 of the physicochemical water quality.
Point 11: Check spelling in line 245.
Response 11: Thank you for the detailed comments of reviewers. We have supplemented as follows this sentence. Thus, the authors have corrected the manuscript (see lines 286-287).
Nevertheless, there was a delicate difference in water quality between normal and rainy season in stagnant conditions.
Point 12: In Figure 5 only 3 species are included, but 5 spp. are mentioned above. maybe you add the temperature to the Graphs also?
Response 12: The authors sincerely appreciate the reviewer's helpful comments. Figure 5 represent of thermal stress results for fish fauna in study site. There are five species of fish in this study, but species not shown in the Figure 5 are tolerant fish species that do not suffer from thermal stress in this study (FPT: Crucian carp 30.0 °C, Minnow 29.0 °C). Thus, the authors have changed the manuscript (see line 355).
Figure 5. Thermal stress results for fish fauna in study site. (a) Open space for stagnant conditions in 2012. (b) Inside the ADP for stagnant conditions in 2012. (c) Open space for circulation conditions in 2013. (d) Inside the ADP for circulation conditions in 2013. Crucian carp and minnow showed ND.
Point 13: Figure 6 / 7: (a) + (b) Change scale --> e.g. 15 to 30. Again only 3 species are included, but 5 spp. are mentioned in each graph. maybe you add the temperature to the Graphs also?
Response 13: The authors sincerely appreciate the reviewer's helpful comments. Figure 5 represent of thermal stress results for fish fauna in study site. There are five species of fish in this study, but species not shown in the Figure 6, 7 are tolerant fish species that do not suffer from thermal stress in this study (FPT: Crucian carp 30.0 °C, Minnow 29.0 °C). Thus, the authors have changed the manuscript (see line 411, line 416).
Figure 6. Diurnal variations in the average water temperature and thermal stress for the fish fauna inhabiting the study site in 2012. (a) Normal season. (b) Rainy season. (1) Thermal stress for various fish fauna in St. 1. (2) Thermal stress for various fish fauna in St. 2. Crucian carp and minnow showed ND.
Figure 7. Diurnal variations in the average water temperature and thermal stress for the fish fauna inhabiting the study site in 2013. (a) Normal season. (b) Rainy season. (1) Thermal stress for various fish fauna in St. 1. (2) Thermal stress for various fish fauna in St. 2. Crucian carp showed ND.
Point 14: What about different size classes / life stages (line 383-386)?
Response 14: The authors sincerely appreciate the reviewer's helpful and insightful comments. Most of the fish covered in this study is for mature fish. However, in general, the juvenile fish is weak and are more vulnerable to the external environment. Therefore, ADP's thermal stress reduction effect is more likely to be beneficial to juvenile fish. These parts should to be addressed in future studies, taking into account the developmental stages of the fish. Thus, the authors have modified the manuscript (see lines 435-436).
Through this analysis, we were able not only to identify when the thermal stress intensively increases, but also to estimate the trend and efficiency of the diurnal of thermal stress on the fish species in the study site. However, because the data were simulated as the mean values of the water temperature, real-time water temperature fluctuations were not fully applied, so the thermal stress of some species may not have been reflected, or may have a lower value. Nevertheless, the diurnal analysis of water temperature and thermal stress are meaningful since the trend of values can be identified over time. Further study is required to investigate the thermal stress in consideration of fish size and life stage based on this study.
Point 15: Where do you suggest to apply a ADP? in a garden pond, in hatcheries --> possible application of your findings need to be discussed (line 391-392 + 411-416)!
Response 15: The authors sincerely appreciate the reviewer's helpful and insightful comments. The pond located in the urban area are widely used in the ecological aspect, but it is easily exposed to the thermal stress due to shallow water depth and shortage of fish shelter. Therefore, if a fish shelter such as ADP is applied to shallow or small aquatic ecosystem (garden or park pond, artificial wetlands etc.), it will be effective in managing the thermal stress of fish. In addition, relatively constant water temperature is maintained inside the ADP, so that the spawning period of a specific fish can be sustained. Thus, the authors have modified the manuscript (see lines 465-468).
Through the experiment, the ADP was found to provide constant and optimal water temperature for living and spawning for bitterling that dominated in the experimental pond during early summer. Also, the ADP provides a more stable fish shelter than open space since the water temperature is maintained at constant levels, thus enabling the fish to avoid the thermal stress induced by the diurnal variations in the water quality. As well as, the ADP can create various environments depending on the depth, and provide various aquaculture environments suitable for different fish species. In particular, as a result of this study, ADP can be applied to shallow water spaces such as garden pond, park pond, and other artificial wetlands because it exhibits effective thermal stress relief of fish in the small aquatic ecosystem. Therefore, the deployment of the ADP appears to offer a practical alternative for effective fishery resources management to improve species diversity and fish communities in a freshwater ecosystem.
Point 16: Try to attract the reader. You should also highlight and discuss the relevance! A detailed discussion on the performance, including a comparison of your designs vs. others in the literature and how the findings can be applied to general scientific community. And finally try to address how the findings can be used in e.g. in a hatchery.
Response 16: The authors sincerely appreciate the reviewer's helpful and insightful comments. Also, the authors fully understand why the reviewer concerned about the sentence for this manuscript. We have supplemented as follows this sentence. Thus, the authors have modified the manuscript (see lines 44-57, lines 465-470).
In this respect, summer, the season of rising temperatures, can be a harsh period for fish. For example, an excessive increase in temperature can induce numerous physiological changes in the body of fish. Previous studies have shown that fish exposed to high temperatures have an adverse effect on protein damage, hormonal changes, and high mortality due to thermal stress [12,13]. Therefore, adequate water management focused on water temperature is closely related to the health of the fish, so the appropriate preparation is required especially in the season when water temperature is rising.
As mentioned, since excessive rises in water temperature in aquatic environments can be detrimental to the reproduction and growth of fish, there has been ongoing research into the functions of fish shelters [14-19,23,24]. Previous studies have found that fish shelters can be used to prevent untimely predator encounters [14,15], to enable the survival of physical disturbance events such as floods or droughts, and to maintain or increase fish populations through enhancing the survival rates [19]. However, most studies have emphasized the complexities of fish shelter structures, and have focused on the vulnerability of prey fish. For this reason, a phenomenological approach is needed to evaluate the function of a fish shelter in mitigating the impact of accumulated thermal stress.
Through the experiment, the ADP was found to provide constant and optimal water temperature for living and spawning for bitterling that dominated in the experimental pond during early summer. Also, the ADP provides a more stable fish shelter than open space since the water temperature is maintained at constant levels, thus enabling the fish to avoid the thermal stress induced by the diurnal variations in the water quality. As well as, the ADP can create various environments depending on the depth, and provide various aquaculture environments suitable for different fish species. In particular, as a result of this study, ADP can be applied to shallow water spaces such as garden pond, park pond, and other artificial wetlands because ADP exhibits effective thermal stress relief of fish in the small aquatic ecosystem. Therefore, the deployment of the ADP appears to offer a practical alternative for effective fishery resources management to improve species diversity and fish communities in a freshwater ecosystem.

Round 2
Reviewer 2 Report
The manuscript has been much improved and is in a nice condition now. I think this manuscript is acceptable.
Reviewer 3 Report
It is improved now, but still needs major language editing. Important is, to include e.g. in line 27 " artificial freshwater ecosystem " And the terminology "stratification" is incorrect (see e.g. https://en.wikipedia.org/wiki/Lake_stratification)Author Response
Point 1: Important is, to include e.g. in line 27 " artificial freshwater ecosystem "
Response 1: The authors sincerely appreciate the reviewer's helpful comments. This study mainly focuses on artificial freshwater areas such as garden pond, park pond, other artificial wetlands, etc. So we have modified the “freshwater ecosystem” to “an artificial freshwater ecosystem” to help readers understand. Thus, the authors have changed the manuscript (see lines 29 of page 1, line 46 of page 16).
Finally, the deployment of the ADP appears to provide a practical alternative for effective fishery resources management to improve species diversity and fish communities in an artificial freshwater ecosystem (garden pond, park pond, other artificial wetlands, etc.).
Therefore, the deployment of the ADP appears to offer a practical alternative for effective fishery resources management to improve the species diversity of fish communities in an artificial freshwater ecosystem.
Point 2: And the terminology "stratification" is incorrect (see e.g. https://en.wikipedia.org/wiki/Lake_stratification)
Response 2: The authors sincerely appreciate the reviewer's helpful and insightful comments. We have tried to improve the readers' understanding by reflecting the point of view of the reviewers by modifying “strafication” to “thermal strafication”(Tadesse et al., 2004; Song et al., 2013). Thus, the authors have changed the manuscript (see lines 21 of page 1, line 17-23 of page 6, line 4 and 14 of page 9, line 31 of page16).
Thermal stratification occurs inside the ADP but is temporarily disturbed due to the mixing from the forced circulation and the rainwater input through rainfall events.
To verify the occurrence of oxic/anoxic layers and thermal stratification inside the ADP, the water temperature, pH, DO, and EC were observed at different depths. As depicted in Figure 4, under stagnant conditions, no significant depth-dependent differences in pH and EC were observed (see Figure 4b,d), whereas both water temperature and DO rapidly decreased at the depth intervals between 0.8 and 1.5 m, indicating that an anoxic layer existed in these water bottoms. In particular, a considerable thermal stratification in average water temperature and DO was observed between depths of 0.8 m and 1.5 m (see Figure 4a,c).
Thus, thermal stratification occurred mainly in the stagnant condition, but not in the circulation condition.
Also, as the water depth increased, the decrease in the water temperature became more pronounced. Overall, the thermal stratification observed in the ADP suggested that different environmental conditions were found depending on the depth. Thus, in terms of water temperature and DO, the ADP can provide various aquaculture environments suitable for different fish species.
In addition, diurnal water temperature fluctuations inside the ADP were steady, but in the open space showed ≥4 °C. Thirdly, thermal stratification occurs but is temporarily disturbed due to the mixing from the forced circulation and the rainwater input through rainfall events.
References
Song, H., Xenopoulos, M.A., Buttle, J.M., Marsalek, J., Wagner, N.D., Pick, F.R., Frost, P.C. 2013. Thermal stratication patterns in urban ponds and their relationships with vertical nutrient gradients. Journal of Environmental Management, 127 317-323.
Tadesse, I., Green, F.B., Puhakka, J.A. 2004. Seasonal and diurnal variations of temperature, pH and dissolved oxygen in advanced integrated wastewater pond systems treating tannery effluent. Water Research, 38, 645-654.
